# Effects of participating in community assets on quality of life and costs of care: longitudinal cohort study of older people in England

Luke Aaron Munford [ID],[1] Anna Wilding,[1] Peter Bower,[2] Matt Sutton[1]

¹Department of Health Organisation, Policy and Economics, Centre for Primary Care and Health Services Research, School of Health Sciences, University of Manchester, Manchester, UK
²NIHR School for Primary Care Research, Centre for Primary Care and Health Services Research, School of Health Sciences, University of Manchester, Manchester, UK

**Correspondence to**
Dr Luke Aaron Munford;
luke.munford@manchester.ac.uk

## ABSTRACT

**Objectives** Improving outcomes for older people with long-term conditions and multimorbidity is a priority. Current policy commits to substantial expansion of social prescribing to community assets, such as charity, voluntary or community groups. We use longitudinal data to add to the limited evidence on whether this is associated with better quality of life or lower costs of care.

**Design** Prospective 18-month cohort survey of self-reported participation in community assets and quality of life linked to administrative care records. Effects of starting and stopping participation estimated using double-robust estimation.

**Setting** Participation in community asset facilities. Costs of primary and secondary care.

**Participants** 4377 older people with long-term conditions.

**Intervention** Participation in community assets.

**Primary and secondary outcome measures** Quality-adjusted life years (QALYs), healthcare costs and social value estimated using net benefits.

**Results** Starting to participate in community assets was associated with a 0.017 (95% CI 0.002 to 0.032) gain in QALYs after 6 months, 0.030 (95% CI 0.005 to 0.054) after 12 months and 0.056 (95% CI 0.017 to 0.094) after 18 months. Cumulative effects on care costs were negative in each time period: £−96 (95% CI £−512 to £321) at 6 months; £−283 (95% CI £−926 to £359) at 12 months; and £−453 (95% CI £−1366 to £461) at 18 months. The net benefit of starting to participate was £1956 (95% CI £209 to £3703) per participant at 18 months. Stopping participation was associated with larger negative impacts of −0.102 (95% CI −0.173 to −0.031) QALYs and £1335.33 (95% CI £112.85 to £2557.81) higher costs after 18 months.

**Conclusions** Participation in community assets by older people with long-term conditions is associated with improved quality of life and reduced costs of care. Sustaining that participation is important because there are considerable health changes associated with stopping. The results support the inclusion of community assets as part of an integrated care model for older patients.

<div style="border:1px solid;">

### Strengths and limitations of this study

► Use of longitudinal cohort data allows us to examine the effects of both starting and stopping participation in community assets.
► Statistical matching strengthens our estimation of the effects of community assets.
► Healthcare costs estimated from linked administrative records.
► Data derived from a single geographical area.
► The estimated effects reflect natural changes in participation in community assets, rather than the effects of a formal social prescribing scheme.

</div>

of healthcare costs in modern economies, and developing innovative ways to deliver cost-effective care for older people with long-term conditions is a policy priority. Although better health and care services are important, they are potentially associated with high costs of delivery and may not be suitable for helping older patients with the challenges they face and the goals they want to achieve. For example, loneliness is prevalent among older patients and may be a significant factor in their health.[1 2] Older patients may prioritise different goals to their healthcare professionals, and those goals (for social support and inclusion, and developing new skills) may be difficult to achieve through conventional health and care services.

In 2010, policy-makers in the UK proposed a 'Big Society',[3] where individuals engaged more with the facilities in their local community, to improve health and well-being through better engagement with 'community assets'. These were defined as '… *the collective resources which individuals and communities have at their disposal, which protect against negative health outcomes and promote health status*',[4] such as charity, voluntary or community groups. Health and social care organisations were advised to support the development and

## INTRODUCTION

Services for managing long-term conditions and multimorbidity are a major component

use of such assets among their populations by mapping community assets and engaging in a process of Asset Based Community Development[5] to help the community increase the health and well-being of its population using activities, skills and assets within the community.

The way in which health and social care organisations engage with community assets has subsequently become more direct. In several areas, health and care professionals (as well as other front-line professionals) have begun to make referrals to such community assets as part of the management of patients, in a process known as 'social prescribing'. Social prescribing has been defined in a number of different ways, but the definition we feel is most appropriate here is '*a mechanism for linking patients with non-medical sources of support within the community*'.[6] It is worth noting here that social prescribing is not limited to patients and is open as a course of action to any individual with a National Health Service (NHS) number. However, we refer to individuals as patients throughout this paper for clarity and consistency.

Social prescribing arrangements are varied across England. In some places, it involves referral, and in others just signposting. In some places, it involves use of existing assets, and in others codesign of new ones. This idea has recently been given new impetus with a commitment in the Long Term Plan for the NHS in England to have over 1000 trained social prescribing link workers in post by 2020/2021 and to expand provision so that over 900 000 people will have been referred to social prescribing schemes by March 2024 (https://www.england.nhs.uk/personalisedcare/social-prescribing/). Within the Long Term Plan, social prescribing is linked to a wider salutogenic model of Universal Personalised Care and seeks to adopt a wider view of care to include a more person-centred model with a focus on well-being and resilience, not just absence of disease.

This rapid expansion of formal provision will occur without a strong evidence base. Although reviews and qualitative work have suggested that community assets improve the health of participants,[7 8] there is limited quantitative evidence.[9] Outcomes that have been identified in qualitative studies have included a sense of involvement and better well-being,[8] whereas outcomes that have been identified in quantitative studies have included health-related quality of life (HRQoL) and healthcare costs.[9] The evidence base for social prescribing is equally limited and has yet to arrive at a consensus.[10] However, it is worth noting that the evidence is still developing in this field, with ongoing qualitative and quantitative studies.

We previously evaluated an integrated care programme for older people that included a programme to improve use of community assets.[9] We used data from a cohort of older people to analyse cross-sectional associations between community asset participation, health and healthcare utilisation. The evidence suggested that community asset participation was associated with significant improvements in health and not significant reductions in healthcare costs. However, the cross-sectional nature of the data meant that we could not interpret the relationships as causal.

In this study, we analyse the relationships between community asset participation, health and healthcare utilisation longitudinally to provide a more rigorous assessment of the causal impact of community asset participation. Using administrative health records further strengthens the analysis presented here as it removes the reliance on recall. As well as considering the uptake of community assets as a possible health enhancing activity, we additionally examine the possibility of there being health decrements associated with ceasing to participate in community assets. A priori, it is not expected that the absolute size of the gains from starting will equal the size of the reductions from stopping.

## METHODS

### Data: cohort description

The data used in this analysis were made available as part of the National Institute of Health Research-funded Comprehensive Longitudinal Assessment of Salford Integrated Care (CLASSIC) study.[11] CLASSIC is an evaluation framework designed to evaluate the Salford Integrated Care Programme (SICP). The SICP is a large-scale integrated care project to transform care for older people with long-term conditions and social care needs. The SICP aims to improve care via a number of mechanisms, including improved access to community assets. Questionnaires were mailed to 12 989 individuals aged 65 years and older with at least one long-term health condition living in the Salford area (a city in the North West of England) between November 2014 and February 2015. These individuals were selected from the disease registers of 33 general practices.

Usable responses were received from 4377 (34%) individuals. These individuals were then sent follow-up questionnaires at 6, 12 and 18 months. At 18 months, responses were revived from 2449 individuals (56% of the baseline cohort). A flow chart showing response rates over time is shown in figure 1.

### Patient and public involvement (PPI)

A study advisory group was formed, whose remit included overseeing management of the entire research project (of which the results presented here are one part), providing a patient voice and commenting on the emerging results and dissemination strategy. We also presented the cohort design and the measures to a local patients group and made changes in response to their feedback. We further presented the cohort design to a local PPI group who provided advice on encouraging people to stay in the cohort.

### Data: variables

#### Health-related quality of life

HRQoL was measured using the Euro-QoL 5D-5L (EQ-5D-5L).[12 13] The EQ-5D-5L is a generic

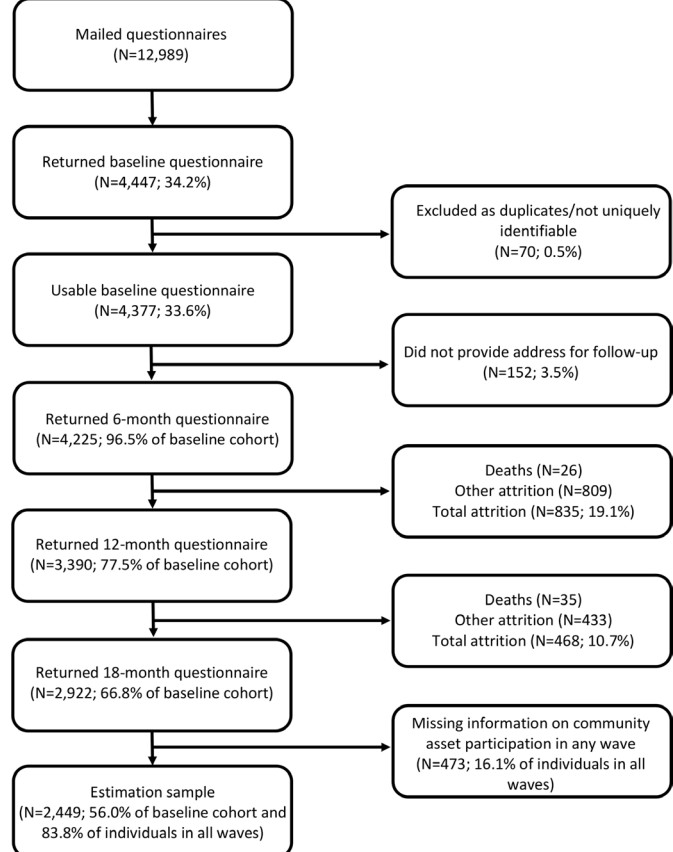

**Figure 1** Description of the cohort.

preference-based measure of HRQoL covering five domains (mobility, self-care, usual activities, pain/discomfort and anxiety/depression).

Participants completed the EQ-5D-5L in the baseline, 6-month, 12-month and 18-month follow-up questionnaires. Responses were converted to a single index utility value based on the crosswalk mapping tool of van Hout *et al*,[14] which maps from the five-level questionnaire onto the three-level questionnaire. This crosswalk tool is the National Institute for Health and Care Excellence (NICE)'s preferred method of obtaining utility values from the EQ-5D-5L.[15] In a robustness check, we used the newly developed algorithm for directly calculating utility scores from the EQ-5D-5L.[16]

Quality-adjusted life years (QALYs) were then calculated at the individual level using the area under the curve method assuming linear extrapolation of utility between time points (Hunter *et al*).[17]

### Healthcare utilisation

Respondents were matched to their administrative health records using NHS Numbers. This allowed us to construct detailed information on use of primary and secondary health services. Individual-level healthcare resource utilisation over the study period was collected from two sources. The number of GP contacts in the previous 6 months was collected from electronic primary care databases. Hospital utilisation was extracted from

linked administrative patient records provided by the NHS, divided into emergency admissions (short stays ≤5, long stays >5 days), elective admissions, elective day cases, outpatient attendances, and accident and emergency (A&E) department attendances, as in Panagioti *et al*.[18]

We costed these activities using NHS Reference Costs, in 2014/2015 values[19] and/or Personal Social Services Research Unit (PSSRU) unit costs.[20] The costs were as follows: elective appointments= £3405; emergency long-stay visits=£2863; emergency short-stay visits=£608; day-case visits=£704; outpatient visits=£112; and visits to A&E=£132.

Information from primary care records contained a count of the number of times an individual visited their GP. We then applied the PSSRU unit cost (in 2014/2015 values) of £65 per visit.[20]

We applied a discount rate of 3.5% to the costs and benefits.[21]

### Net benefit

As in our earlier work,[9] we defined net benefits as an individual's QALY gain minus the cost of their healthcare utilisation.[22] We used the two thresholds used by the NICE, namely £20 000 and £30 000 but focus mainly on the £20 000 threshold for reasons of brevity.

### Community asset participation

Community asset participation was defined as a binary variable equal to 1 if an individual reported participating in any one of a list of activities, and 0 otherwise. The list of community assets is included in online supplementary appendix table A1, along with reported participation rates over time.

### Demographic and socioeconomic characteristics

We controlled for gender and age using a series of 5-year age categories (ranging from 65 to 69 years, up to 85+ years). The reference age group is 65–69 years. We also controlled for living situation, coded as 'live with spouse', 'live with other' or the reference category 'live alone'. We included binary variables for each of the following qualifications: 'one or more Ordinary Level (O-Levels)/ Certificate of Secondary Education (CSEs)/General Certificate of Secondary Education (GCSEs)', 'one or more A-Levels/AS-Levels', 'Degree', 'National Vocational Qualification (NVQ)', 'Trade qualifications', 'Professional qualifications'). An individual can tick multiple responses. The reference category was 'no qualifications'. The variables used in this analysis are summarised in online supplementary appendix table A2.

### Statistical methods

We used double-robust estimation[23] to estimate the impact of community asset participation on: (1) HRQoL, (2) costs of formal healthcare services and (3) net social benefit.[22]

Double-robust estimation is a form of treatment effects estimator that accounts for observable factors that could influence treatment. The method combines a propensity

score model with a regression adjustment. The propensity score is obtained from a logistic regression of community asset participation on baseline covariates. The inverse of this propensity score is then used to weight the regression model for the outcome.[23] As long as one model is correctly specified, the double-robust estimator produces unbiased results.[24 25] If both models are correctly specified, then double-robust estimator is both unbiased and efficient.[26]

The choice of control variables for both models is important. We provide a full list of all variables included in both the treatment (propensity score) equation and the outcome (regression adjustment) model in an online supplementary appendix table A2.

Analysis was performed in Stata (V.15.1). Double-robust estimation was implemented using the *teffects ipwra* command, which by default assumes a linear model in the outcome equation.

### Primary analysis

Our primary analysis focuses on the individuals who provided information on their participation in community assets in all four waves of the survey. To assess if initial community asset participation was associated with whether the respondent remained in the sample, we ran a logistic model of drop-out as a function of baseline characteristics, including health and community asset participation. We interacted baseline community asset participation with all the covariates to see if there were differential associations of drop-out with the covariates between those who did or did not participate in community assets at baseline.

### Uptake analysis

For the 6-month analysis, we defined the comparator group as those individuals who did not participate in community assets at baseline and continued to not participate at the 6-month follow-up. The treatment group consists of those individuals who did not participate in assets at baseline but did report participation at 6 months. This is comparison A (table 1).

For the 12-month and 18-month analyses, the definition of the treatment group was more complicated. As there are three time points in the 12-month analysis and four time points in the 18-month analysis, there are $2^3=8$ and $2^4=16$ different possible combinations of participation and non-participation, respectively. We focused on the 'best case scenario' in the primary analyses.

In the 12-month and 18-month analyses, the comparator group is those individuals who never participated (NNN or NNNN). The primary definition of treatment in the 12-month analysis was NYY (comparison C) and in the 18-month analysis was NYYY (comparison E).

### Cessation analysis

We followed a similar logic for estimating the effects of ceasing to participate in community assets. For the 6-month analysis, we defined the comparator group as

**Table 1** List of comparison groups and definitions of control and treatment groups

| Comparison | Pattern of community asset participation | |
| --- | --- | --- |
| | Control group | Treated group |
| A: 6-month uptake analysis | NN | NY |
| B: uptake sensitivity analysis | NNN | N?Y |
| C: 12-month uptake analysis | NNN | NYY |
| D: uptake sensitivity analysis | NNNN | N??Y |
| E: 18-month uptake analysis | NNNN | NYYY |
| F: 6-month cessation analysis | YY | YN |
| G: cessation sensitivity analysis | YYY | Y?N |
| H: 12-month cessation analysis | YYY | YNN |
| I: cessation sensitivity analysis | YYYY | Y??N |
| J: 18-month cessation analysis | YYYY | YNNN |

Y indicates participation. N indicates non-participation. ? indicates either participation or non-participation.

those who always participate and the treatment group as those individuals who initially participated at baseline and then stopped by the 6-month follow-up, comparison F. The 12-month and 18-month analyses followed a similar pattern and are shown as comparisons H and J in table 1.

### Secondary analyses

In a secondary analysis, we relaxed the restriction that an individual had to remain in the sample for all four waves. We included data from all individuals in the respective waves.

In another secondary analysis, we additionally considered the effects of participating in community assets at the 12-month or 18-month follow-up, regardless of what happened in the interim periods. For the uptake analysis, these were comparisons B and D in table 1. For the cessation analysis, these were comparisons G and I.

### RESULTS

Selected characteristics of the respondents at baseline are available in table 2. Further detail is provided in online supplementary appendix table A2.

### Participation in community assets over time

Figure 2 shows how many people participated in community assets at each wave.

Participation in community assets increased over time (table 2). The largest increase in participation occurred between baseline (53%) and the 6-month follow-up (57%). Mean levels of HRQoL decreased over time for both participants and non-participants.

### Attrition analysis

The only significant predictors of drop-out from the cohort were older age and education. However, the

**Table 2** Changes over time in health-related quality of life, costs of healthcare utilisation, participation and selected baseline summary statistics by initial participation status

| | Pooled (n=2449) | Initial non-participants (n=1146) | Initial participants (n=1303) |
|---|---|---|---|
| **EQ5D scores over time** | | | |
| EQ5D score (BL) | 0.759 (0.234) | 0.712 (0.263) | 0.792 (0.204) |
| EQ5D score (FU6) | 0.752 (0.238) | 0.705 (0.268) | 0.791 (0.202) |
| EQ5D score (FU12) | 0.751 (0.239) | 0.704 (0.270) | 0.792 (0.199) |
| EQ5D score (FU18) | 0.742 (0.239) | 0.699 (0.268) | 0.784 (0.207) |
| **Healthcare costs over time** | | | |
| Healthcare costs (−6 to B) | 1661.73 (2072.78) | 1779.89 (2231.93) | 1557.71 (1916.64) |
| Healthcare costs (B to FU6) | 1754.97 (2063.16) | 1850.86 (2204.30) | 1670.52 (1927.28) |
| Healthcare costs (FU6 to FU12) | 1489.33 (1730.47) | 1519.78 (1815.86) | 1463.06 (1651.90) |
| Healthcare costs (FU12 to FU18) | 2347.15 (2512.30) | 2476.51 (2789.90) | 2233.26 (2234.53) |
| **Participation rates over time (%)** | | | |
| CA participation rate (B) | 53 | 0 | 100 |
| CA participation rate (FU6) | 57 | 24 | 86 |
| CA participation rate (FU12) | 58 | 24 | 87 |
| CA participation rate (FU18) | 59 | 28 | 87 |
| **Selected covariates at baseline** | | | |
| Female | 0.52 | 0.52 | 0.54 |
| Aged 65–69 years | 0.32 | 0.32 | 0.31 |
| Aged 70–74 years | 0.28 | 0.27 | 0.29 |
| Aged 75–79 years | 0.21 | 0.21 | 0.22 |
| Aged 80–84 years | 0.12 | 0.13 | 0.11 |
| Aged 85+ years | 0.07 | 0.08 | 0.06 |
| Live alone | 0.35 | 0.35 | 0.34 |
| Live with spouse | 0.59 | 0.58 | 0.61 |
| Live with other | 0.06 | 0.07 | 0.05 |
| No qualifications | 0.42 | 0.52 | 0.35 |
| School level qualifications | 0.28 | 0.17 | 0.37 |
| College level qualifications | 0.1 | 0.05 | 0.15 |
| University level qualifications | 0.07 | 0.05 | 0.1 |
| NVQ and trade qualifications | 0.23 | 0.22 | 0.24 |
| Professional qualifications | 0.22 | 0.16 | 0.26 |

For continuous outcomes, SD are given in parentheses.

FU6 is the 6-month follow-up.

BL, baseline; CA, community asset; EQ5D, EuroQol 5 Dimension; FU, follow-up period; NVQ, National Vocational Qualification.

magnitude of their effects on drop-out were not significantly different between those who initially participated and those who initially did not participate in community assets. The full regression results are presented in online supplementary appendix table A3.

### Statistical tests of suitability of the propensity score

Online supplementary appendix figure A1 shows the distributions of the propensity scores before and after matching. Panel A shows the distributions for the uptake analysis, and panel B shows the distributions for the cessation analysis. In both cases, the matching considerably improves the similarity between the control and treatment groups.

### Multivariate analysis: uptake analysis

There is a positive and statistically significant effect of starting community asset participation on HRQoL (table 3, panel A). The benefit of starting to participate in community assets is a 0.017 QALY gain (95% CI 0.002 to 0.032) compared with those who never participate in assets at the 6-month follow-up. The effect of starting to

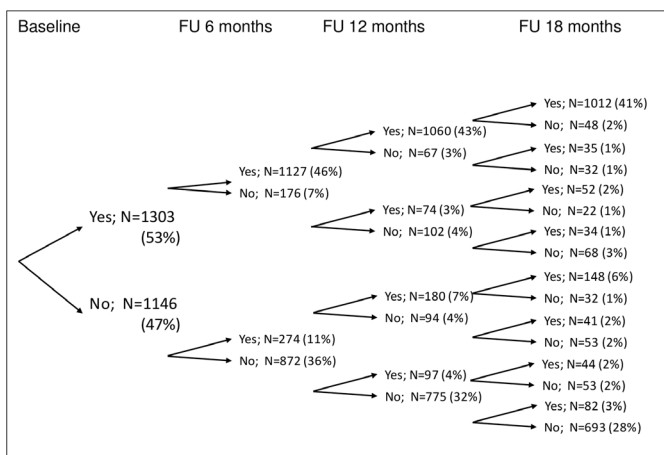

Baseline FU 6 months FU 12 months FU 18 months

**Figure 2** Longitudinal patterns of community asset participation. The percentages in the final column may not sum to 100 due to rounding.

participate in community assets is a QALY gain of 0.030 (95% CI 0.005 to 0.054) at the 12-month follow-up and a QALY gain of 0.056 (95% CI 0.017 to 0.094) at 18 months.

Starting to participate in community assets reduced costs in the 6-month period by £96 (95% CI £–512 to £321), in the 12-month period by £283 (95% CI £–926 to £359) and in the 18-month period by £453 (95% CI

£–1366 to £461). While these effects are in the direction expected, they are not statistically significant.

Assuming a willingness-to-pay of £20 000 per QALY, the 6-month net benefit of starting to participate in community assets was £155 per participant (95% CI £13 to £297). The 12-month net benefit was £734 per participant (95% CI £66 to £1403) and the 18-month net benefit was £1956 per participant (95% CI £209 to £3703).

### Multivariate analysis: cessation analysis

When we consider cessation (table 3, panel B), we found that stopping participating in community assets led to a QALY decrease of 0.036 at the 6-month follow-up (95% CI–0.068 to –0.004). The corresponding QALY losses for the 12-month and 18-month follow-ups were 0.068 (95% CI –0.132 to –0.005) and 0.102 (95% CI –0.173 to –0.031), respectively.

When we considered the total costs of healthcare utilisation, we found that stopping participating in community assets led to large and statistically significant increases in healthcare utilisation costs. In the 6-month period, this increase was £689 (95% CI £162 to £1216), whereas in the 12-month and 18-month follow-ups, these increases were £857 (95% CI £252 to £1463) and £1335 (95% CI £113 to £2558), respectively.

| Table 3 | The effect of starting community asset participation on outcomes | | |
|---|---|---|---|
| | **(1)** | **(2)** | **(3)** |
| | **QALYs** | **Total cumulative cost in£s** | **Net benefit in £s (based on £20 000 per-annum)** |
| Panel A: uptake analysis | | | |
| BL versus FU6 | 0.017 (0.002 to 0.032), p=0.022 | −95.59 (−511.84 to 320.65), p=0.653 | 154.74 (12.56 to 297.22), p=0.033 |
| BL versus FU12 | 0.030 (0.005 to 0.054), p=0.019 | −283.42 (−925.50 to 358.66), p=0.387 | 734.27 (66.02 to 1402.53), p=0.031 |
| BL versus FU18 | 0.056 (0.017 to 0.094), p=0.004 | −452.56 (−1365.89 to 460.74), p=0.331) | 1955.50 (208.50 to 3702.50), p=0.028 |
| Panel B: cessation analysis | | | |
| BL versus FU6 | −0.036 (−0.068 to −0.004), p=0.029 | 689.00 (161.69 to 1216.31), p=0.010 | −624.35 (−1224.21 to −24.50), p=0.041 |
| BL versus FU12 | −0.068 (−0.132 to −0.005), p=0.034 | 857.27 (251.68 to 1462.86), p=0.006 | −1653·42 (−2959.04 to −347.79), p=0.013 |
| BL versus FU18 | −0.102 (−0.173 to −0.031), p=0.005 | 1335·33 (112.85 to 2557.81), p=0.032 | −3894·42 (−7256.51 to −532.33), p=0.023 |

Variables in the outcome equation: gender, age (in 5-year groups), living arrangements, employment status, education and presence of limiting conditions. Variables in the matching equation: gender, age (in 5-year groups), living arrangements, employment status, education, presence of limiting conditions, satisfied with transport, EQ5D domains scores (not utility value), the ICEpop CAPability measure for Older people (ICECAP-O) score, six questions from the Social Support Inventory, distance to nearest community asset and cost of healthcare services in previous 6 months (before baseline). Net benefit calculations assume a threshold value of £20 000 per annum (hence £10 000 per 6 months and £30 000 for 18 months). In the uptake analysis, BL versus 6 months compares NN (control group) with NY (treatment group). BL versus 12 months compares NNN (control group) with NYY (treatment group). BL versus FU18 compares NNNN (control group) with NYYY (treatment group). In the cessation analysis, BL versus 6 months compares YY (control group) with YN (treatment group). BL versus 12 months compares YYY (control group) with YNN (treatment group). BL versus FU18 compares YYYY (control group) with YNNN (treatment group).
FU6 is the 6-month follow-up.
BL, baseline; FU, follow-up period; QALYs, quality-adjusted life years.

Additionally, there were negative net benefits (assuming a £20 000 NICE threshold) associated with cessation. In the 6-month period, this potential loss was £624 per participant per year (95% CI £–112 to £–25), whereas in the 12-month and 18-month follow-up periods, this loss was £1653 per participant per year (95% CI £–2959 to £–348) and £3894 per participant per year (95% CI £–7257 to £–532), respectively.

## Secondary analyses

The results using all available data on respondents are qualitatively similar in terms of magnitude and statistical significance (online supplementary appendix table A4).

Use of less strict definitions of uptake and cessation also produces similar results, but the effects are typically smaller in magnitude (online supplementary appendix table A5).

## DISCUSSION

Our study involved a large sample of patients recruited and followed up over an 18-month period. Although there was loss to follow-up, the overall rate of retention was reasonable. We collected detailed data on asset use and health, with objective data on healthcare costs available from administrative records. We adopted rigorous methods for the estimation of causal effects and found the main results were robust to several assumptions.

We additionally performed many robustness/sensitivity analyses where we changed the variables included in the matching model. Our main results remained qualitatively similar in all cases, and we concluded that our main findings were not sensitive to the choice of variables used in the matching equation.

However, the study was conducted in a single region in England, in a population of older people living in an area undergoing transformation of older people's services. Care must therefore be taken in generalising from this context. According to Public Health England, Salford is among the 20% most deprived districts in England with lower life expectancy than the national average. Ninety-four per cent of residents are white. However, Salford has experienced many healthcare reforms in the recent past, particularly around older people. As a result, Salford is the first 'age Friendly City' and the Age Well campaign has experienced considerable success. The SICP programme also ensured that there was more integration of care within Salford, particularly during the study period. Therefore, the results need to be interpreted in this context, where there has been significant investment in community assets locally.

As we highlighted in previous work, objective data on the impact of increasing use of community assets is limited,[9] and this paper therefore makes a significant contribution to this area. Our broad results are consistent with the published work in this field while adding value due to the methodological strengths of the work.

Haslam et al[27] undertook a longitudinal study of the relationship between engagement with social groups and cognitive function using data from the English Longitudinal Study of Ageing (ELSA). They found that current use of social groups significantly predicted better cognition. Their study differs from ours in that we are interested in health and healthcare utilisation, and we model the decision to partake in social groups and community assets.

Also using ELSA, Steffens et al[28] analysed the relationship between social group participation and quality of life and mortality, particularly around the time of retirement. They showed that engagement with social groups led to better quality of life and a reduced risk of premature death. They used a 'matched control group' approach and had a much smaller treated sample. We argue that the methods used here, as well as the wider suite of outcome measures, reinforces their message that starting to use community assets and social groups can significantly improve health.

Two analyses by Cruwys et al have considered the relationship between social group participation and depression.[29 30] They show, using various data sources, that membership of more clubs was associated with a lower probability of future depression and that identification with a social group predicts recovery from depression. Our results are consistent with this in that depression has been shown to be a major driver of HRQoL[31] and healthcare utilisation.[32]

Social prescribing schemes play a key role in the NHS Long Term Plan. Although popular with services and policy makers, a recent review of such schemes found significant issues with the quality of the evidence base,[10] with only 2 of the 15 evaluations having any sort of comparator. This evidence base is continually evolving, and we expect this to change given a number of ongoing and planned evaluations.

Our analytical methods provided a comparator group to better assess the impact of changes in asset use. We examined non-experimental changes in asset use in the context of a wider integrated care initiative, which saw some patients starting to use assets, and others ceasing use. It is plausible that at least some of this increased use reflected the wider integrated care initiative that was being undertaken in the area, but this cannot be determined reliably. Our analysis used a large sample and robust analytic methods and was able to assess the effects of starting and stopping asset use. However, we were not testing the impact of new referrals to community assets, and we cannot be sure that the benefits of the changes we assessed would necessarily translate to patients in formal social prescribing schemes. Nevertheless, our results make an important contribution, given the policy interest in these approaches and the limited evidence base.

Our results highlight that the effects of starting and stopping asset use are not symmetrical, which suggests that equal attention needs to be given to these different processes. The focus of social prescribing tends to be on the former, but our data suggest that it is important to

identify people whose use of assets stops. If such people can be identified and supported, the gains might be even greater, but it is not clear that the same schemes would be suited for increasing use and maintaining use.

### Unanswered questions and future research

As noted previously, the study was conducted in a single region of England, and the results would need replication. Given that the benefits of asset use seemed to increase with time, further long-term evaluation would also be indicated. Exploration of the reasons why people stop using assets, and whether it can be identified and managed more effectively, would also be a research priority. Since this study was completed, two further schemes have been launched in the surrounding areas: one in Salford, the Well-being Matters scheme; and one in Greater Manchester, the Person Centred and Community Approaches scheme. These schemes were launched in December 2018 and March 2019, respectively, and might provide the basis for future research in this area.

Another potential limitation is that we do not observe the timing of events. For example, in the cessation analysis, we know that individuals ceased participation in community assets and they experience a decline in QALYs. We assume that the former caused the latter, but it may be possible that declining HRQoL led to a cessation in asset participation. The statistical matching on baseline characteristics should somewhat mitigate against this if we assume that initial levels of HRQoL and health indicate similar rates of decline, conditional on age and other factors. However, without detailed dates of when community asset participation stopped, we cannot be certain of the sequence of events.

In our analysis, we are unsure if individuals chose to start (or stop) using community assets because they were referred to them by a link worker (a social prescriber), or if they chose to do so for other reasons (including friend referrals, more exposure and so on). Therefore, while we demonstrate that community assets have considerable benefits, we cannot be completely confident that this is all attributable to social prescribing.

Furthermore, we cannot confidently demonstrate which type of community assets are most beneficial, as our definition of utilisation is based on self-reports.

Our results provide a robust assessment of the impacts of changes in the use of community assets and provide further impetus to calls for robust evaluation of their effects. There is a legitimate debate as to whether the standard controlled trial is optimal for the assessment of such schemes, given their flexible nature (and the importance of patient choice) and the likely impact of context (include local availability of assets) that may complicate evaluation, although there are examples of evaluation using trial methodology.[33]

## CONCLUSION

We used quasiexperimental methods to explore the impact of changing patterns of the use of community assets in a population of older people living in an area that introduced an integrated care initiative that sought to increase asset use.

We found that increasing use of community assets was associated with increased HRQoL, reduced costs and positive societal net benefit. The reduction in costs and positive net benefits were sustained over time and indicated substantial benefits from prolonged community asset use.

The effects of starting to use assets were not symmetrical to those from ceasing use, with the latter associated with larger losses. This is important, as encouraging use among those who do not currently use assets may require different policy and patient-level interventions to those designed to encourage continued use.

The results support the inclusion of community assets as part of an integrated care model for older patients.

**Acknowledgements** We would like to thank North West e-Health and the National Institute for Health Research (NIHR) Clinical Research Network: Greater Manchester for assistance with the recruitment of the Comprehensive Longitudinal Assessment of Salford Integrated Care (CLASSIC) cohort, as well as staff at the participating practices. For assistance with the CLASSIC study, we would like to thank 'Salford Together'—a partnership of Salford City Council, National Health Service (NHS) Salford Clinical Commissioning Group, Salford Royal NHS Foundation Trust, Greater Manchester Mental Health NHS Foundation Trust and Salford Primary Care Together.

**Contributors** PB, MS and LAM made substantial contributions to the design of the study. All authors contributed to analysis and interpretation of the data. LAM drafted the paper, and PB, MS and AW all revised drafts. All authors gave final approval for the version to be published and are accountable for the integrity of the work.

**Funding** Funding was provided by the UK NIHR (grant 12/130/33). LAM was supported by the Medical Research Council, through a Skills Development Fellowship (grant number MR/N015126/1).

**Disclaimer** Views and opinions are those of the authors and do not necessarily reflect those of the NHS, NIHR, NIHR Evaluation, Trials and Studies Coordinating Centre, Health Services and Delivery Research, Medical Research Council or Department of Health.

**Competing interests** None declared.

**Patient consent for publication** Not required.

**Ethics approval** Ethics approval was obtained from the National Research Ethics Service North West Lancaster (Research Ethics Committee reference 14/NW/0206).

**Provenance and peer review** Not commissioned; externally peer reviewed.

**Data availability statement** Data are available on reasonable request. Data may be obtained from a third party and are not publicly available. The data that support the findings of this study are available from the principal investigator of the original study but restrictions apply to the availability of these data, which were used under licence for the current study and so are not publicly available. Data are however available from the authors upon reasonable request and with permission.

**ORCID iD**
Luke Aaron Munford http://orcid.org/0000-0003-4540-6744

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
