## [Reviewer comments · BMJ Open]

ARTICLE DETAILS

TITLE (PROVISIONAL)	EFFECTS OF PARTICIPATING IN COMMUNITY ASSETS ON QUALITY OF LIFE AND COSTS OF CARE: LONGITUDINAL COHORT STUDY OF OLDER PEOPLE IN ENGLAND
AUTHORS	Munford, Luke; Wilding, Anna; Bower, Peter; Sutton, Matt

VERSION 1 – REVIEW

REVIEWER	Rachel Sumner University of Gloucestershire, United Kingdom
REVIEW RETURNED	21-Aug-2019

GENERAL COMMENTS	The study sets out to examine whether participation in community-based assets has positive outcomes for older people. The study has an important story for social prescribing literature, as it is adding much needed information with regard to QoL and demand for health care service utilisation. The large sample size, and longitudinal approach of this study, along with its economic assessment, are extremely useful additions to the field and will be of interest internationally as social prescribing increases. The report is written extremely well, with an excellent level of clarity and detail. I have very little in the way of recommendations for improvement for this paper, but have provided some thoughts below that mostly cover the precision of language and the potential to explore the data further. The authors note (p.4) that “This is defined as ‘enabling healthcare professionals to refer patients to a link worker, to co-design a nonclinical social prescription to improve their health and well-being’”. I agree that this is a definition of social prescribing, however the term itself is rather loosely used in practice (rightly or wrongly) to encompass both this rather formal definition, and other routes as well. For example, in some areas of the UK, social prescribing is done more directly, where link workers (or community connectors, health connectors – the list goes on...) are not necessarily required in the process. Here, patients can be referred directly into programmes by GPs, nurses, social care workers, and social prescribers. Frequently, there is no co-design involved in these prescriptions either, as they rely on existing community asset infrastructures. I guess what is important here, to me, would be that social prescribing currently has many guises. The term is also often used (incorrectly, in my opinion) for models that effectively just provide signposting. Importantly for the present work, it is not always about a bespoke service for patients, but is about utilising existing community assets, which is where I think the authors are heading with the present work. The Chatterjee paper provides a broad definition sourced from the CentreForum, which may be more apt for the context of the paper: “Social prescribing is defined as: “A mechanism for linking patients with non-medical sources of support
---

	within the community” (CentreForum Metal Health Commission, 2014, p. 6). On p8, the authors note: “we applied a discount rate of 3.5% to the costs and benefits”, with a citation for the NICE document “Guide to the methods of technology appraisal 2013”. Could the authors expand on the significance of this discount please? Summary statistics are presented in an appendix table (A2). As summary statistics are not presented elsewhere, it would be good to see the N and % of categorical variables (as opposed to mean values) to provide a description of the sample. P12 line 19 – minor typo “Assuming a willingness to pay of £20,000 per QALY” Please provide ethical approval details. Is it possible to provide some overview of SES of the cohort? For example, using index of multiple deprivation? This might add an interesting perspective to the data, as existing larger studies that look at social prescribing typically tend to involve cohorts from areas of lower deprivation. It is currently unclear where/how the element of multimorbidity fits into these data and their analyses. Beyond the assumption that those of this age cohort would be multimorbid, I have not seen any specific outlining of how multimorbidity is ascertained or incorporated into the overall methodology of the analyses. Beyond the title, abstract, and “strengths and limitations” the term multimorbidity is not used again. P.6 outlines that surveys were mailed to “individuals... with at least one long-term health condition...”, but how is multimorbidity (which could be argued to be conceptually slightly larger than the diagnosis of more than one long-term condition) defined? It is also possible, given the definition provided here, that there are individuals with only one long-term condition, which would be indicative of not having multimorbidity. P. 15, line 2 – minor typo “They show, using various data sources, that that” In the discussion the authors cite the Bickerdike review paper to evidence that literature in the field of social prescribing presents several issues. I would agree with this, however there have been papers produced since this review that address some of the issues presented in that review, such as lack of substantial sample sizes and robust analyses. I don’t think they need to be described or cited, but I would request that this statement be revised to bring a representation of the field more up to date by stating that evidence is developing (or similar). I do agree with the authors’ point about studies using comparators; although in most literature concerning social prescribing, having control groups is not usually possible. I would actually argue that the only way a comparator group could be used for analysis of the activities used in SP is with the methodology employed in this paper.
--	---

REVIEWER	Dr Michelle Howarth University of Salford School of Health and Society Frederick Road Campus
-----------------	--

	Salford M6 6PU UK
REVIEW RETURNED	21-Aug-2019

GENERAL COMMENTS	General Comments This is an interesting and very timely paper, particularly following the publication of the NHSE Long Terms plan and the roll out of the PCCA and social prescribing within emerging PCN's. That said, the authors need to ensure that relevant contemporary developments are included in the paper, which will add context and strengthen the implications of this paper for commissioners going forward. I have really enjoyed reading this paper and consider this paper to be of value to readers, commissioners and services. There are a few comments below, plus suggested amendments required before this paper should be accepted. Page 3: Box – strengths & limitations of the study – rather than state that longitudinal data allowed us to....suggest amend to 'longitudinal data enabled examination'..... Page 4: Line 23: correction needed – it's not just health and social care professionals that can refer – this needs to be amended to all front-line professionals as per NHSE guidance and National Social Prescribing Framework. Page 5: Social prescribing is one aspect of the NHSE Comprehensive Model of Personalised Care – the authors should not divorce the process of social prescribing from this significant context. Recognition of the way in which this movement promotes salutogenic as opposed to merely pathogenic principles is an important aspect of NHSE Universal Personalised Care – this detail needs to be included in the paper for this to be meaningful to professionals, services and commissioners. Page 5: The authors state that they have previously identified that ABCD approach resulted in significant improvements- please include an example of the outcome. Page 6: Line 10: ?typo should be 'at' rather than 'and' Page 7: Suggest that health care utilisation is optimum for older people who are more likely to live with co-morbid conditions – hence, outpatient appointments rarely reduce due to external factors – as they are needed to sustain wellbeing. Was this considered as a variable in the analyses? The authors suggest that they evaluated GP consultations/contact – but were these face to face or telephone appointments – or were these contact with GPs ie through the receptionists – was the type of consultation captured? The link worker role is key to the social prescribing referral process – the link worker will have invariably held a wellbeing conversation with the older person – was this considered? What type of community assets were included? These range considerably from 12 week therapeutic horticulture programmes through to knit and natter groups – how was this accounted for in the analysis? The discussion section needs to be developed further – the authors note that this is one area and others may struggle to generalise from this, however it must also be noted that Salford is the first 'age Friendly City' and the Age Well campaign has experienced considerable success. This context needs to be included within the paper. Page 15: line 2: 'That' is repeated X2 The discussion should not remove social prescribing from the personalised care agenda. Significant investment and commitment
---

	has been provided across GM through the PCCA and as such, should be discussed within the start and discussion of the paper so that implications for policy and practice are transparent.
--	--

REVIEWER	Catherine Best University of Stirling, UK
REVIEW RETURNED	26-Sep-2019

GENERAL COMMENTS	Thank you for asking me to review this manuscript. I think it could make an important contribution. My comments to the authors are as follows The study has found that starting to participate in community assets is associated with improved health related quality of life. The reverse was also demonstrated that is, that stopping participation was associated with greater reductions in health related quality of life. The most significant potential confounder in these relationships are post baseline changes in health. For example, if a participant's health deteriorates post-baseline, for example they experience a stroke, then they are likely to reduce community participation and to report lower health related quality of life at follow up. As I understand it, health was used as a baseline predictor but changes in health conditions were not modelled as predictors of outcomes. In this complex data set I understand it may not be possible to conduct such analyses but this must be acknowledged as a limitation on the conclusions that can be drawn from the current study. The use of doubly robust estimation relies on an assumption that treatment is independent of outcome conditional on the observed variables. The authors need to provide more information on how this was assessed. The decision to include co-morbid conditions as a set of 23 binary variables each representing the limiting effects of one health condition needs further justification. The sample was selected as having co-morbid conditions. Therefore, there is likely to be covariance amongst these variables. Using the variables in this way suggests you wish to control for the effects of particular health problems conditional on the effects of other health problems. I cannot see the rationale for this. It would be more useful to include a single measure characterising the degree of functional impairment from all conditions. To a certain extent this will be measured by the HRQoL measure but there are more sensitive measures available. This and the use of the Social Support questions as separate variables increases the dimensionality of the data and reduces the model degrees of freedom. The purpose of including as separate variables needs further justification. More information is need on the analysis. What software was used? What distributional assumptions were made- e.g. did the model for QALY outcomes use an identity link and normal distribution for example? The ICECAP-0 is included in A2 as a treatment equation variable but is not listed as one in the table 3 footnotes. The paper is clearly written and well presented. Further information on the analyses as described above, will improve the manuscript.
---

VERSION 1 – AUTHOR RESPONSE

Reviewer: 1

Reviewer Name: Rachel Sumner

Institution and Country: University of Gloucestershire, United Kingdom

Please state any competing interests or state 'None declared': None declared.

The study sets out to examine whether participation in community-based assets has positive outcomes for older people. The study has an important story for social prescribing literature, as it is adding much needed information with regard to QoL and demand for health care service utilisation. The large sample size, and longitudinal approach of this study, along with its economic assessment, are extremely useful additions to the field and will be of interest internationally as social prescribing increases. The report is written extremely well, with an excellent level of clarity and detail. I have very little in the way of recommendations for improvement for this paper, but have provided some thoughts below that mostly cover the precision of language and the potential to explore the data further.

We thank the reviewer for their kind words regarding our manuscript, as well as their detailed and thoughtful suggestions and comments.

The authors note (p.4) that "This is defined as 'enabling healthcare professionals to refer patients to a link worker, to co-design a nonclinical social prescription to improve their health and well-being". I agree that this is a definition of social prescribing, however the term itself is rather loosely used in practice (rightly or wrongly) to encompass both this rather formal definition, and other routes as well. For example, in some areas of the UK, social prescribing is done more directly, where link workers (or community connectors, health connectors – the list goes on...) are not necessarily required in the process. Here, patients can be referred directly into programmes by GPs, nurses, social care workers, and social prescribers. Frequently, there is no co-design involved in these prescriptions either, as they rely on existing community asset infrastructures. I guess what is important here, to me, would be that social prescribing currently has many guises. The term is also often used (incorrectly, in my opinion) for models that effectively just provide signposting. Importantly for the present work, it is not always about a bespoke service for patients, but is about utilising existing community assets, which is where I think the authors are heading with the present work. The Chatterjee paper provides a broad definition sourced from the CentreForum, which may be more apt for the context of the paper: "Social prescribing is defined as: "A mechanism for linking patients with non-medical sources of support within the community"" (CentreForum Mental Health Commission, 2014, p. 6).

We thank the reviewer for pointing this reference out, and agree it is more appropriate for our context. As such, we have modified the text and now use this reference.

On p8, the authors note: "we applied a discount rate of 3.5% to the costs and benefits", with a citation for the NICE document "Guide to the methods of technology appraisal 2013". Could the authors expand on the significance of this discount please?

This discount rate is standard in NICE technology appraisals. The rationale is that costs and benefits incurred today are usually valued more highly than costs and benefits occurring in the future. Discounting health benefits reflects society's preference for benefits to be experienced in the present rather than the future. Discounting costs reflects society's preference for costs to be experienced in the future rather than the present. Further detail is available at <https://www.nice.org.uk/process/pmg9/chapter/glossary#discounting-2> and https://assets.publishing.service.gov.uk/government/uploads/system/uploads/attachment_data/file/712699/tag-unit-a1.1-cost-benefit-analysis-may-18.pdf

Summary statistics are presented in an appendix table (A2). As summary statistics are not presented elsewhere, it would be good to see the N and % of categorical variables (as opposed to mean values) to provide a description of the sample.

Descriptive statistics for the key variables of interest are presented in Table 2 in the main manuscript. In Table A2, the mean for the categorical variables is the proportions for which the statement is true (or the dummy variable in 'on'). For example, in the first row the 'sex' variable having a mean of 0.52 means that 52% of the sample is female. We do not present the % in each response for all categorical variables for reasons of space as we condition on a large set of variables. We have

added a row at the end explaining that all summary statistics are for the complete case sample, where N=2,449.

P12 line 19 – minor typo “Assuming a willingness to pay of £20,000 per QALY”

We have now hyphenated ‘willingness-to-pay’.

Please provide ethical approval details.

Ethical approval details is provided in the ‘Declarations’ section (page 19).

Is it possible to provide some overview of SES of the cohort? For example, using index of multiple deprivation? This might add an interesting perspective to the data, as existing larger studies that look at social prescribing typically tend to involve cohorts from areas of lower deprivation.

We thank the reviewer (as well as reviewer 2) for pointing this out. As a result, we have now added some more context on Salford to the discussion section, outlining that it is relatively deprived and predominantly white. However, Salford is seen as somewhat of a success story for improving the experiences of ageing people.

It is currently unclear where/how the element of multimorbidity fits into these data and their analyses. Beyond the assumption that those of this age cohort would be multimorbid, I have not seen any specific outlining of how multimorbidity is ascertained or incorporated into the overall methodology of the analyses. Beyond the title, abstract, and “strengths and limitations” the term multimorbidity is not used again. P.6 outlines that surveys were mailed to “individuals... with at least one long-term health condition...”, but how is multimorbidity (which could be argued to be conceptually slightly larger than the diagnosis of more than one long-term condition) defined? It is also possible, given the definition provided here, that there are individuals with only one long-term condition, which would be indicative of not having multimorbidity.

We thank the reviewer for pointing this out. We agree with the reviewer, and as such have deleted ‘multimorbidity’ from the new title.

P. 15, line 2 – minor typo “They show, using various data sources, that that”

The second ‘that’ has been deleted.

In the discussion the authors cite the Bickerdike review paper to evidence that literature in the field of social prescribing presents several issues. I would agree with this, however there have been papers produced since this review that address some of the issues presented in that review, such as lack of substantial sample sizes and robust analyses. I don’t think they need to be described or cited, but I would request that this statement be revised to bring a representation of the field more up to date by stating that evidence is developing (or similar). I do agree with the authors’ point about studies using comparators; although in most literature concerning social prescribing, having control groups is not usually possible. I would actually argue that the only way a comparator group could be used for analysis of the activities used in SP is with the methodology employed in this paper.

Thank you for pointing this out. We have now updated the text in the introduction (p5) and discussion (p15) sections to acknowledge that this evidence base is still developing.

Reviewer: 2

Reviewer Name: Dr Michelle Howarth

Institution and Country:

University of Salford

School of Health and Society

Frederick Road Campus

**Salford
M6 6PU
UK**

Please state any competing interests or state 'None declared': None declared

Please leave your comments for the authors below

General Comments

This is an interesting and very timely paper, particularly following the publication of the NHSE Long Terms plan and the roll out of the PCCA and social prescribing within emerging PCN's. That said, the authors need to ensure that relevant contemporary developments are included in the paper, which will add context and strengthen the implications of this paper for commissioners going forward. I have really enjoyed reading this paper and consider this paper to be of value to readers, commissioners and services. There are a few comments below, plus suggested amendments required before this paper should be accepted.

We thank the reviewer for their kind words and helpful suggestions.

Page 3: Box – strengths & limitations of the study – rather than state that longitudinal data allowed us to.....suggest amend to 'longitudinal data enabled examination'.....

The strengths and limitations box has now been edited in its entirety, as per the suggestions of the editor.

Page 4: Line 23: correction needed – it's not just health and social care professionals that can refer – this needs to be amended to all front-line professionals as per NHSE guidance and National Social Prescribing Framework.

Thank you for pointing this out. We have added "as well as other front-line professionals" to the text to make this clear and emphasise it is not just health and social care professionals who can refer. We have also modified our definition of social prescribing, as per reviewer 1's point.

Page 5: Social prescribing is one aspect of the NHSE Comprehensive Model of Personalised Care – the authors should not divorce the process of social prescribing from this significant context. Recognition of the way in which this movement promotes salutogenic as opposed to merely pathogenic principles is an important aspect of NHSE Universal Personalised Care – this detail needs to be included in the paper for this to be meaningful to professionals, services and commissioners.

We thank the reviewer for highlighting this. We have now added some text to the introduction (p5) articulating that Social Prescribing is one element of a bigger model of care.

Page 5: The authors state that they have previously identified that ABCD approach resulted in significant improvements- please include an example of the outcome.

We have included examples of outcomes from both the qualitative and quantitative literatures (p5).

Page 6: Line 10: ?typo should be 'at' rather than 'and'

This has now been changed to 'at'.

Page 7: Suggest that health care utilisation is optimum for older people who are more likely to live with co-morbid conditions – hence, outpatient appointments rarely reduce due to external factors – as they are needed to sustain wellbeing. Was this considered as a variable in the analyses?

We did not consider specific types of utilisation as outcomes, but instead costed the 'whole package'. We did this as we did not want to pick up substitution effects. We are interested in the cost of health care services used, not specific types of services. For example, we do not want to pick up effects of

changing from outpatient to inpatient care, where by definition one would decrease and one would increase. We believe that considering the overall utilisation allows us to abstract away from this issue.

Additionally, as a robustness check (which we do not include in the paper) we re-estimated the models only considering expenditure on emergency care. The results reduce slightly in magnitude, but the same story holds: starting participation in community assets leads to a societal net-benefit and stopping participation leads to a societal net-loss.

The authors suggest that they evaluated GP consultations/contact – but were these face to face or telephone appointments – or were these contact with GPs ie through the receptionists – was the type of consultation captured?

We included both face-to-face and telephone consultations.

The link worker role is key to the social prescribing referral process – the link worker will have invariably held a wellbeing conversation with the older person – was this considered?

No, we cannot account for this as we do not know the exact reason why people start to use (or stop using) community assets. Information in medical records did not contain social prescription, and individuals were not asked in the cohort questionnaire. We have added text to this effect in the 'Unanswered questions and future research' subsection of the discussion.

What type of community assets were included? These range considerably from 12 week therapeutic horticulture programmes through to knit and natter groups – how was this accounted for in the analysis?

The seven broad categories of community assets that we considered are listed in Table A1 in the supplementary appendix. Utilisation was self-reported into categories by the respondents. As above, we have included a short paragraph in the 'Unanswered questions and future research' subsection explaining this in the manuscript.

The discussion section needs to be developed further – the authors note that this is one area and others may struggle to generalise from this, however it must also be noted that Salford is the first 'age Friendly City' and the Age Well campaign has experienced considerable success. This context needs to be included within the paper.

Thank you for pointing this out, particularly around the recent successes Salford has experienced with regards to 'ageing well'. As per reviewer 1's comment, we have now added some more context on Salford to the discussion section, outlining that it is relatively deprived and predominantly white. However, Salford is seen as somewhat of a success story for improving the experiences of ageing people.

Page 15: line 2: 'That' is repeated X2

The second 'that' has been deleted.

The discussion should not remove social prescribing from the personalised care agenda. Significant investment and commitment has been provided across GM through the PCCA and as such, should be discussed within the start and discussion of the paper so that implications for policy and practice are transparent.

This is a good point, and as such we have added some text to the 'Data: cohort description' section explaining that the CLASSIC programme was part of a bigger integrated care programme. This is again highlighted in the discussion section, where we now emphasise that these results should be interpreted in this context, where there had already been considerable investment of community assets locally.

Reviewer: 3

Reviewer Name: Catherine Best

Institution and Country: University of Stirling, UK

Please state any competing interests or state 'None declared': None declared

Thank you for asking me to review this manuscript. I think it could make an important contribution. My comments to the authors are as follows

Thank you for your positive comments, we respond in turn below. We believe they improve the quality of the manuscript.

The study has found that starting to participate in community assets is associated with improved health related quality of life. The reverse was also demonstrated that is, that stopping participation was associated with greater reductions in health related quality of life. The most significant potential confounder in these relationships are post baseline changes in health. For example, if a participant's health deteriorates post-baseline, for example they experience a stroke, then they are likely to reduce community participation and to report lower health related quality of life at follow up. As I understand it, health was used as a baseline predictor but changes in health conditions were not modelled as predictors of outcomes. In this complex data set I understand it may not be possible to conduct such analyses but this must be acknowledged as a limitation on the conclusions that can be drawn from the current study.

We thank the reviewer for pointing this out. We have added some text to the Discussion section (pp16-17) explaining this.

"Another potential limitation is that we do not observe the timing of events. For example, in the cessation analysis we know that individuals ceased participation in community assets and they experienced a decline in QALYs. We assume that the former caused the latter, but it may be possible that declining HRQoL led to a cessation in asset participation. The statistical matching on baseline characteristics should somewhat mitigate against this if we assume that initial levels of HRQoL and health indicate similar rates of decline, conditional on age and other factors. However, without detailed dates of when community asset participation stopped, we cannot be certain of the sequence of events."

The use of doubly robust estimation relies on an assumption that treatment is independent of outcome conditional on the observed variables. The authors need to provide more information on how this was assessed.

The doubly-robust estimation is more flexible than other 'treatment effects' estimations as it will produce unbiased estimates of the treatment effect if either the allocation into treatment equation (decision to start or stop using community assets) or the outcome equations (the effects of community asset on health) are correctly specified.

Exact tests for the 'treatment independent of outcomes' assumption are not available. However, we implemented the Oster (2016) method and showed that the main results (the treatment effects) are not sensitive to the choice of observable we include. We therefore conclude that we have a good set of observable (informed by previous literature and theory), and that given the robustness of the results to removing observables, they should also be robust to omitted unobservable variables (which may or may not be independent of the outcome).

The decision to include co-morbid conditions as a set of 23 binary variables each representing the limiting effects of one health condition needs further justification. The sample was selected as having co-morbid conditions. Therefore, there is likely to be covariance amongst these variables. Using the variables in this way suggests you wish to control for the effects of particular health problems conditional on the effects of other health problems. I cannot see the rationale for this. It

would be more useful to include a single measure characterising the degree of functional impairment from all conditions. To a certain extent this will be measured by the HRQoL measure but there are more sensitive measures available. This and the use of the Social Support questions as separate variables increases the dimensionality of the data and reduces the model degrees of freedom. The purpose of including as separate variables needs further justification.

We have run analyses using the count of limiting conditions instead, and the model fit is slightly reduced. However, the main coefficients of interest (the treatment effects) remain qualitatively unchanged – they have very similar magnitudes and levels of statistical significance (or precision). We cannot reject the hypothesis that the two effects (including the conditions separately or as a composite measure) are the same. We therefore include each condition separately in the main manuscript as we believe that adding a composite measure (such as the count of conditions that limit daily activity) may mask much variability (e.g. having three limiting conditions could mean a large number of possible combinations).

More information is need on the analysis. What software was used? What distributional assumptions were made- e.g. did the model for QALY outcomes use an identity link and normal distribution for example?

We have added some text in the ‘Statistical methods’ section explaining this.

“Analysis was performed in Stata (version 15.1). Double-robust estimation was implemented using the *teffects ipwra* command, which assumes a linear model in the outcome equation.”

We are aware of a strand of literature that finds that in certain scenarios, allowing for a beta distribution provides a better fit for modelling QALYs. These scenarios typically involve a mass-point towards the top of the distribution (1 in HRQoL measures; some other maximum value in other outcomes). However, at this point in time, the double robust estimates are not able to allow for beta distributions. They can handle count outcomes (Poisson) and binary outcomes (probit and logit), but unfortunately not truncated and/or beta distributions. We have estimated the models manually allowing for a beta distribution in the outcome equation. We obtain very similar treatment effects, but estimated with different levels of precision. This is due to not being able to account for the two-stage process fully when estimating these equations manually. However, we take some comfort from the fact that the treatment effects are similar under normal and beta distributions.

The ICECAP-0 is included in A2 as a treatment equation variable but is not listed as one in the table 3 footnotes.

Thank you for pointing this out. This has been added to the footnote in Table 3.

The paper is clearly written and well presented. Further information on the analyses as described above, will improve the manuscript.

Thank you again for your comments, we agree that they have improved the manuscript.

References

Oster, E. (2016) Unobservable Selection and Coefficient Stability: Theory and Evidence, *Journal of Business & Economic Statistics*, 37(2), 187-204; Available at: <https://www.tandfonline.com/doi/full/10.1080/07350015.2016.1227711>

VERSION 2 – REVIEW

REVIEWER	Rachel Sumner University of Gloucestershire
----------	--

REVIEW RETURNED	27-Nov-2019
GENERAL COMMENTS	The authors have done a commendable job in attending to the various recommendations by the reviewers. This is an excellent piece of research that will undoubtedly provide a significant contribution to the field of social prescribing. I recommend the article be accepted for publication.
REVIEWER	Dr Michelle Howarth University of Salford School of Health & Society Salford Manchester M6 6PU
REVIEW RETURNED	26-Nov-2019
GENERAL COMMENTS	Thank you for this revised submission. This remains a very timely paper, and one which will certainly support the social prescribe movement. The authors have addressed the key points, and there are some very minor changes required which I have listed below. These are mainly contextual, which I hope will strengthen the paper and ensure meaning and relevance.  1. General proof read is needed (I noticed a couple of typos, additional full stops etc) 2. The abstract conclusion is a little ambiguous, could this be re-worded? 3. The definition of social prescribing is improved, but the authors need to note that it is not always 'patients' that are referred. In Salford, referrals through the community connector scheme can originate from any health or social care professional, as long as an NHS number is provided. 4. Page 5, line 9...change of date is needed – according to the NHSE Long Term Plan, Link Workers will be in every GP practice by 2020....not 2021 5. Page 5 line 20...authors needs to re-word 'quantities' to 'quantitative'. 6. Evidence citation for page 5 lines 10-13 needed 7. The authors have based their paper on Salford....and therefore may need to include the Wellbeing Matters Social Prescribing programme in the discussion as this play a key role in supporting ABCD. Similarly, the authors may like to reference the Greater Manchester PCCA model – so that readers are aware of the significant influences that have enabled the ABCD approach.
REVIEWER	Catherine Best University of Stirling
REVIEW RETURNED	06-Dec-2019
GENERAL COMMENTS	Thank you for giving the opportunity to review this paper. I believe the authors have addressed the majority of comments. I think perhaps that the causal language in the abstract could be adjusted. In the conclusion 'negative effects' could be replaced. Doubly robust estimation could be the basis for causal statements but only in the absence of unobserved confounding either one of the treatment or outcome model. Post baseline changes in health status could potentially confound both models so I think caution in describing these effects as causal is warranted.

VERSION 2 – AUTHOR RESPONSE

Reviewer: 2

Reviewer Name: Dr Michelle Howarth

Thank you for this revised submission. This remains a very timely paper, and one which will certainly support the social prescribe movement. The authors have addressed the key points, and there are some very minor changes required which I have listed below. These are mainly contextual, which I hope will strengthen the paper and ensure meaning and relevance.

1. General proof read is needed (I noticed a couple of typos, additional full stops etc)

We have now proof read the document and removed the double full-stop from the abstract. We have also made minor grammatical changes elsewhere.

2. The abstract conclusion is a little ambiguous, could this be re-worded?

In line with Reviewer 3's comment, we have amended the conclusions section of the abstract to read:

“Participation in community assets by older people with long-term conditions is associated with improved quality of life and reduced costs of care. Sustaining that participation is important because there are considerable health changes associated with stopping. The results support the inclusion of community assets as part of an integrated care model for older patients.”

3. The definition of social prescribing is improved, but the authors need to note that it is not always ‘patients’ that are referred. In Salford, referrals through the community connector scheme can originate from any health or social care professional, as long as an NHS number is provided.

Thank you for pointing this out. We have added two sentences on page 5 explaining this in more detail. *“It is worth noting here that social prescribing is not limited to patients, and is open as a course of action to any individual with an NHS number. However, we refer to individuals as patients throughout this paper for clarity and consistency.”*

4. Page 5, line 9...change of date is needed – according to the NHSE Long Term Plan, Link Workers will be in every GP practice by 2020....not 2021

Thank you for pointing this out. We have amended this to “...link workers in post by 2020/2021...” in line with the Long Term Plan (<https://www.england.nhs.uk/personalisedcare/social-prescribing/>)

5. Page 5 line 20...authors needs to re-word ‘quantities’ to ‘quantitative’.

This has now been corrected.

6. Evidence citation for page 5 lines 10-13 needed

The reference to the social prescribing element of the Long Term Plan has now been included as a weblink on page 5.

7. The authors have based their paper on Salford....and therefore may need to include the Wellbeing Matters Social Prescribing programme in the discussion as this play a key role in supporting ABCD. Similarly, the authors may like to reference the Greater Manchester PCCA model – so that readers are aware of the significant influences that have enabled the ABCD approach.

The Wellbeing Matters scheme (part of Salford Together) was launched in December, 2018 and hence comes after our analysis period (<https://www.salfordtogether.com/2018/12/wellbeing-matters/>). Therefore, we do not mention in the manuscript, but do recognise it is a very valuable local scheme. In fact, this study can be thought of as highlighting the good that Wellbeing Matters can do, particularly within Salford. Likewise, the Greater Manchester PCCA was designed and started to be implemented after the analysis period considered in this analysis (March 2019; <https://www.gmcvo.org.uk/HSCEngage/PCCA>, with some work done before this, but not before 2016).

Reviewer: 1

Reviewer Name: Rachel Sumner

The authors have done a commendable job in attending to the various recommendations by the reviewers. This is an excellent piece of research that will undoubtedly provide a significant contribution to the field of social prescribing. I recommend the article be accepted for publication.

We thank the reviewer for their kind words, and valuable comments on suggestions on the earlier version of the paper.

Reviewer: 3

Reviewer Name: Catherine Best

Thank you for giving the opportunity to review this paper. I believe the authors have addressed the majority of comments.

Response

I think perhaps that the causal language in the abstract could be adjusted. In the conclusion 'negative effects' could be replaced. Doubly robust estimation could be the basis for causal statements but only in the absence of unobserved confounding either one of the treatment or outcome model. Post baseline changes in health status could potentially confound both models so I think caution in describing these effects as causal is warranted.

We have reworded the conclusions section of the abstract, by replacing “*negative effects*” with “*considerable health changes associated with stopping*”. Replacing ‘effects’ with ‘associated with’ tones down the causal language.